# Health Risk Modifiers of Exposure to Persistent Pollutants among Indigenous Peoples of Chukotka

**DOI:** 10.3390/ijerph17010128

**Published:** 2019-12-23

**Authors:** Valery Chashchin, Aleksandr A. Kovshov, Yngvar Thomassen, Tatiana Sorokina, Sergey A. Gorbanev, Boris Morgunov, Andrey B. Gudkov, Maksim Chashchin, Natalia V. Sturlis, Anna Trofimova, Jon Ø. Odland, Evert Nieboer

**Affiliations:** 1Arctic Biomonitoring Laboratory, Northern (Arctic) Federal University, 163002 Arkhangelsk, Russia; valerych05@mail.ru (V.C.); Yngvar.Thomassen@stami.no (Y.T.); t.sorokina@narfu.ru (T.S.); gudkovab@nsmu.ru (A.B.G.); megaarctic@list.ru (N.V.S.); a.trofimova@narfu.ru (A.T.); 2Northwest Public Health Research Center, 191031 St. Petersburg, Russia; nfo@s-znc.ru (A.A.K.); info@s-znc.ru (S.A.G.); 3Laboratory of Arctic Medicine, Mechnikov Northwestern State Medical University, 191015 St. Petersburg, Russia; rectorat@szgmu.ru; 4National Research University Higher School of Economics, 101000 Moscow, Russia; bmorgunov@hse.ru; 5Norwegian University of Life Sciences, 0033 Oslo, Norway; 6Department of Hygiene and Human Ecology, Northern State Medical University, 163002 Arkhangelsk, Russia; 7NTNU, The Norwegian University of Science and Technology, 7491 Trondheim, Norway; 8Department of Biochemistry and Biomedical Sciences, McMaster University, Hamilton, ON L8S 4K1, Canada; nieboere@mcmaster.ca

**Keywords:** persistent contaminants, Chukotka, indigenous peoples, health risk assessment, risk modifying factors

## Abstract

The aim of the study was to assess temporal trends in health risks related to most common persistent contaminants, including polychlorinated biphenyls (PCBs), dichloro-diphenyl-trichloroethanes (DDTs), lead (Pb), as well as mercury (Hg) among indigenous peoples living in coastal areas of Chukotka in Arctic Russia. This is examined in relation to exposure pathways and a range of social and behavioral factors capable of modifying the exposure to these contaminants, including place of residence, income, traditional subsistence, alcohol consumption, and awareness of risk prevention. The primary exposure pathway for PCBs is shown to be the intake of traditional foods, which explained as much as 90% of the total health risk calculated employing established risk guidelines. Nearly 50% of past DDT-related health risks also appear to have been contributed by contaminated indoor surfaces involving commonly used DDT-containing insecticides. Individuals who practiced traditional activities are shown to have experienced a 4.4-fold higher risk of exposure to PCBs and a 1.3-fold higher risk for DDTs, Pb, and Hg. Low income, high consumption of marine mammal fat, alcohol consumption, and lack of awareness of health risk prevention are attributed to a further 2- to 6-fold increase in the risk of PCBs exposure. Low socioeconomic status enhances the health risks associated with exposure to the persistent contaminants examined.

## 1. Introduction

Traditional subsistence in Arctic indigenous communities can be disrupted by factors such as severe changes in climate, uncontrolled environmental pollution, and contamination of traditional foods [1,2,3,4]. Abuse of tobacco, alcohol, and the remoteness of the Arctic communities further complicate matters. Multiple studies in European and North America Arctic areas have demonstrated that individuals subsisting on traditional foods are found to have higher concentrations of environmental pollutants in their bodies, as do breastfed children [5].

There is ample knowledge that negative socioeconomic and behavioral factors enhance the prevalence of health impairments that reduce life expectancy, increase disease incidence rates and premature mortality among the indigenous communities in Arctic Russia [6]. In this context, quantitative studies have not been conducted in relation to the health risks associated with different pathways of exposure to persistent contaminants such as organochlorines (OCs) and toxic metals, nor of the modifying effects of socioeconomic and behavioral factors on the exposure intensity to these environmental contaminants.

The international efforts of the Arctic Monitoring and Assessment Programme (AMAP) in surveying the health status of indigenous communities residing in the Far North in the context of exposure to environmental contaminants have generated regular reports beginning in the late 1990s (e.g., [7,8,9]). A series of measures to reduce the risk of harmful anthropogenic impacts have also been implemented. Nevertheless, the actual socioeconomic impact of the rehabilitation actions has remained underexplored. However, for health risk mitigation to be effective the general recommendations made by AMAP in their 2001 and 2004 projects, namely safe waste disposal methods and dietary recommendations, proved to be insufficient.

### Objectives

The study aimed to assess the health risks among the indigenous people of Chukotka and focuses on pathway-specific exposures for the most common persistent contaminants in order to explore in some detail the impact of common socioeconomic and behavioral factors on cancer and non-cancer risks.

## 2. Methods

### 2.1. Study Design

The study targeted randomly selected groups of indigenous individuals who were permanent residents of the coastal village of Uelen and the inland village of Kanchalan in the Chukotka Autonomous District (Okrug) in the Far East Region of Russia (for map see [10]). In the 2001 study, 251 residents of Uelen aged between 18 and 71 years were enrolled, of whom 132 were women. A 2010 follow-up included 86 individuals aged 27–67, of whom 48 were women. The 2001 study also included residents of Kanchalan considered to be a referent group to the coastal population because of large differences in traditional diets and occupations [7,10]. This component consisted of 360 individuals aged between 19 and 81, of whom 208 were women. Informed consent was obtained from all of the study participants. The field work involved the administration of a questionnaire. The study protocol, lifestyle, the dietary questionnaire, and sampling strategy adopted were approved by the Human Health Expert Group established by the AMAP Secretariat at the Svalbard meeting, Norway, 6–10 May 2001. The study protocol was also approved by the Ethical Committee of the Pasteur Institute, St Petersburg (international code # T5096). For consistency, the same study protocol and questionnaire surveys were employed in both the 2001 and 2010 surveys to facilitate comparisons in the exposure to persistent contaminants and related health effects experienced by the indigenous people.

The structure of the database consists of 500 informational fields (boxes) completed in strict compliance with the AMAP questionnaire [8]. The data comprises social and economic status, lifestyle details, education, occupational risk factors, dietary details and habits, personal use of the materials potentially contaminated by toxic metals and/or POPs, health history, outcomes of the latest medical examination, and the analytical results for concentrations of metals in whole blood and POPs and lipids in serum. More details on the social and economic status of study populations and contaminant exposure patterns available in the Appendix A posted in the websites (links are provided below).

Chemical Analysis of Contaminants

The chemical analyses of the environmental and blood samples for both communities were made by the internationally accredited laboratory located in St. Petersburg, namely the Northwest Branch of Research and Production Association referred to as “Typhoon” employing validated analytical procedures approved by AMAP [11] Our focus in the current publication is on mercury (Hg), lead (Pb), dichloro-diphenyl-trichloroethanes (DDTs), and polychlorinated biphenyls (PCBs). The sum of DDT and its metabolites included 4,4′-DDE, 4,4′-DDD, 4,4′-DDT, 2,4-DDE, 2,4-DDD, and 2,4-DDT; while the sum of PCBs included 15 congeners, namely 28/31, 52, 99, 101, 105, 118, 128, 138, 153, 156, 70, 180, 183, and 187.

### 2.2. Heath Risk Assessment

The health risk assessments presented in this article are limited to those contaminants measured in blood of our study subjects at concentrations close to or exceeding their acceptable concentrations in blood (whole or serum), namely: PCBs, DDTs, Pb, and Hg.

In order to quantify the noncarcinogenic risk of PCBs intake through atmospheric air, potable water (orally), household surface scrapings, soil dust (oral and inhaled), as well as traditional and grocery store foods, we used reference concentrations, tentative maximum permissible concentrations and levels prescribed by the Russian statutory acts and instruments (see below). In addition, pertinent data on PCB concentrations in food stuffs, the environment, and household indoor surfaces were employed in the calculations as described in [8]

The impacts on contaminant body burdens of occupation (a focus on traditional subsistence or other occupations) and remoteness of the two selected communities were examined. Note that only men are involved in the traditional subsistence trades (hunting, reindeer herding, fishing). In addition to occupation, the influences of the following social status indicators were explored: monetary income level per capita (below or above the regional living wage), alcohol and marine mammal fat intake, and an awareness of the prevention measures for exposure to persistent contaminants. The study protocol and written consent form were approved by the Local Committee for Biomedical Ethics of the Northwest Public Health Research Center, St. Petersburg.

Based on the participant-reported quantities of foods consumed, health risks pertaining to the exposure were calculated for PCBs, DDT pesticides, Pb, and Hg using the USA Environmental Protection Agency (USA EPA) Guidance R 2.1.10.1920-04 for Human Health Risk Assessment from Environmental Chemicals [12]. To quantify the intake of PCBs and DDTs from contact with contaminated indoor household surfaces, the methods of the USA EPA were employed [13,14,15]. In calculating reference doses (i.e., recommended acceptable daily intake of persistent contaminant) for associated noncarcinogenic outcomes the hazard quotient (HQ) was selected, as was the cancer slope factor (CSF) for quantifying lifetime carcinogenic risks (CRs). We were guided by the up-to-date information available from the USA EPA, the OEHHA (The U.S. Office of Environmental Health Hazard Assessment), and AMAP. The CSFs used for dietary intake were PCBs, 2 mg/kg/day; DDTs, 0.34 mg/kg/day [12]; and Pb, 0.0083 mg/kg/day [14]; and the reference doses for the recommended acceptable daily intake of contaminants without appreciable health risk were: PCBs, 0.0003 mg/kg/day; DDT, 0.05 mg/kg/day; Pb, 0.00357 mg/kg/day; and Hg, 0.00071 mg/kg/day [8].

In order to quantify the harmful impact based on biomarkers of exposure, we adopted a model for converting the PCB concentrations (aggregate of 15 congeners) in blood plasma (µg/L, coefficient of determination = 0.600) into the daily average total PCB intake. This model relied on the linear regression equation: x = (y – c)/B = (y – 1.789)/0.031; with the lower limit of a 95% confidence interval of x = (y – 0.011)/0.018; an upper limit of a 95% confidence interval: x = (y – 3.567)/0.045, with y equal to the PCB concentration in blood plasma (in µg/L), B the nonstandardized coefficient (PCB); c, a constant; and x the PCB daily intake (ng/kg/day). Hg and Pb blood concentrations were similarly converted into estimates of their daily intakes. Measurement of PCBs in plasma and of Hg and Pb in whole blood met AMAP’s stringent quality control measures and those of the Norwegian Institute of Occupational Health [8,9].

### 2.3. Statistical Methods

The data was processed using the common statistical, computer-based tools that included Microsoft (Excel 2013; Microsoft Corporation, Redmond, WA, USA) and dedicated software (IBM SPSS Statistics v. 22, SPSS Inc., Chicago, IL, USA) and employed the Mann–Whitney–Wilcoxon rank sum test, and independent/dependent samples T test. For evaluation of normalcy of distribution, the Kolmogorov–Smirnov test was used. The critical significance value of the null hypothesis was taken as 0.05, and the calculated risk levels are cited with their 95% confidence intervals.

## 3. Results

The 2001 data in Table 1 indicate that the calculated CRs associated with the intake of PCBs relates primarily (84.2%) to the consumption of traditional foods that included sea fish (chum salmon, humpback salmon, char) and especially sea mammals (walrus and seal meat and fat). By comparison the pathway-specific CR calculations for the 2010 survey indicated little change (76.0%) in the relative risks of PCB exposure (Table 2), and thus traditional food remained the core contributor to the aggregate risk.

The relative carcinogenic/noncarcinogenic health risk ratios for DDTs and PCBs based on the 2001 data in Table 1 for Uelen are 6.75 and 380, respectively. Furthermore, it is clear that the dominating exposure pathways were the consumption of traditional foods and contact with indoor household surfaces in both 2001 (Table 1) and in 2010 (Table 2).

The analysis of the dietary intake of PCBs, DDTs, and mercury (data of 2001) indicates that a number of statistically significant differences existed between the population of Uelen and that of Kanchalan in terms of the intakes of PCBs, DDTs, and mercury, and the health risks they are associated with (Table 3). The coastal population of Uelen experienced six-fold higher health risks in relation to exposures to PCBs (*p* < 0.001) than the continental population of Kanchalan. A comparable pattern is seen for the risks of exposure to DDTs: in Uelen they were around five times higher *p* < 0,001). Interestingly, it has been reported that a 1.9-fold increase in the risk of exposure to DDTs has been found among reindeer herders because of the use of anti-insect treatment of farmed reindeer. For Hg, both types of risk were three-fold higher in Uelen residents than in Kanchalan (*p* < 0.001). When comparing the exposure related to traditional subsistence activity for males with exposure in other occupations in Uelen, Pb and Hg intakes were about 30% higher for the former by comparison, both CR and HQ health risks associated with PCBs exposures were ≈4.5-fold higher for males engaged in traditional subsistence gathering than in other occupations. In Uelen, the total carcinogenic risk associated with Pb exposure was small and is estimated as 9.7 × 10^−7^ [7.2 × 10^−7^ − 2.9 × 10^−6^] and for noncarcinogenic outcomes it was 0.03 [0.02–0.10]. Note that the International Agency for Research on Cancer (IARC) designates Pb as a group 2A carcinogen [16]. In Kanchalan the corresponding values were 1.3 × 10^−6^ [0.0–2.7 × 10^−6^] and 0.04 [0.0–0.09], respectively.

The calculated PCB-related health risks in the population of Uelen in 2010 are shown in Table 4. It is evident that low monetary income level per capita, high consumption of marine mammal fat, abuse of alcohol, traditional subsistence, and lacking awareness of measures to prevent exposure to contaminants are responsible for a 1.8- to 5.7-fold higher risk of exposure.

## 4. Discussion

Our study has shown that the health risks due to the potential exposure to persistent contaminants are the highest in the coastal population of Uelen when compared to the more inland community of Kanchalan. For males practicing traditional subsistence (namely, hunting, fishing, reindeer herding), the persistent contaminant-induced health risks largely relate to PCBs, whereas the conclusions for DDT, Pb, and Hg are less certain due to the significant overlap of concentration confidence intervals in the sample analyzed.

The increased noncarcinogenic risk of exposure of the coastal population of Chukotka to PCBs is primarily due to a decrease in immunity [3]. The health outcomes associated with PCB exposure include enhanced risks of infectious diseases (including tuberculosis see Figure 1 in [3]); respiratory diseases; pathological processes in endocrine and urogenital systems; congenital anomalies (impaired development); and adverse pregnancy outcomes [8]. The enhanced noncarcinogenic risks observed in the population of Uelen are is deemed unacceptable. It is caused mainly by the intake of PCB by way of traditional food.

We have previously identified [1] that social factors other than occupation and place of residence can impact toxicological risk levels and the current analyses of the 2010 data for Uelen confirm this.

Despite the focus on improving sanitation in the early 2000s, the majority of the indigenous residents remain unaware of the risks relating to the exposure to persistent contaminants and its prevention. This is reflected in the absence of a reduction in related risks. Unfortunately, low income prevents the purchase of costly foods from grocery stores as and up to 90% of the diet still involves fish and, in the coastal areas, sea mammals as well. The latter continue to be a prominent source of persistent contaminants [5].

Alcohol abuse remains high among the indigenous population and home brewing is common. In one respect, alcohol is an obstacle to receiving quality education and well-paid jobs, and forces the indigenous communities to subsist on the traditional foods. Consumption of home-brewed beer remains a special health risk as it often involves the use of the containers previously used such as technical liquid storage tanks (e.g., for other fuels). This practice may have contributed to the 3.4-fold increase in the risk of PCB exposure.

It is evident that traditional food is a common intake pathway for persistent contaminants. Out of all the economic industries in Chukotka, traditional subsistence is the least profitable [4], which explains why indigenous people cannot afford to consume grocery store foods in sufficient quantities to reduce contaminant intake. Exposure during the preparation of self-made Pb pellets is another issue, as well as the inhalation of Pb fumes released during gun use. Furthermore, the consumption of meat in which Pb fragments from the Pb shot are embedded is another well-established source [17]. Furthermore, the increased risk of DDT exposure among reindeer herders has contributions from the past use of DDT-based insecticides to protect reindeer from blood-sucking insects.

Apart from traditional food consumption, the risk of harmful exposure to PCB and DDT has contributions from contaminated household surfaces that appear to have been routinely treated/contaminated with DDT-based insecticides and indoor use of PCB-containing technical oils (estimated to contain up to 51% of DDTs and 14% of PCBs, respectively). Both factors are difficult to control. There are no provisions in the existing legal framework in Russia to specify the contents of persistent contaminants in the fish and sea mammals consumed; nor that transferred from surfaces that come into contact with foods. Moreover, even if there were such provisions and the associated norms, they would hardly be of any help to regulate the actual consumption and pre-consumption treatment of food stuffs. In this context, highly relevant is the development of healthy diet recommendations for High North (arctic) residents. Nevertheless, their development would be enhanced if complemented by educational measures and increases in income among indigenous peoples, as well as addressing the accumulated environmental damage. The lack of vitamins, mineral salts, and microelement intake can be resolved by imported foods or supplements. For these to be affordable, new employment needs to be generated, as well as improvements in and access to vocational education. Additionally, improved sanitation practices should be promoted that enhance the indigenous community-friendly context.

The conventional approach to risk assessment focuses on doses of the harmful substances suitable for noncarcinogenic and/or carcinogenic risk calculation purposes [2]. However, for such calculations to be reliable they need to involve not only laboratory analyses of harmful substances in foods and pertinent environmental samples. Detailed investigation of the exposure pathways is also essential. It may be difficult for ordinary residents to estimate their daily food consumption rates, especially since specific studies provide data on daily rates within a limited time period. For this reason, we deem it most expedient that health risk assessment should at first be based on biomarkers of human exposure, as this allows for a focus on individuals. Concrete recommendations on the reduction of health risks can then be initiated based on the primary sources identified in published studies (e.g., the extensive AMAP database (e.g., [7,8])). This approach may, for example, prove useful in planning for a pregnancy as excessive concentrations of persistent contaminants (in both females and males) have been reported to be associated with higher risk of pregnancy complications, miscarriages, congenital anomalies, and impaired development, whereas prolonged breastfeeding causes the mother’s fat deposits to lose lipophilic persistent contaminants and the baby’s body to accumulate them (e.g., [18]). 

## 5. Conclusions

The primary uptake pathway for prominent carcinogenic and noncarcinogenic environmental contaminants for the indigenous population of Chukotka is through the consumption of traditional subsistence foods, the contribution of which to the overall health risk profile may score as high as 90%. The exposure to DDTs does not only derive from the consumption of locally caught fish and sea mammals, because secondary contamination of food stuffs from contact with various household surfaces constitutes an additional source.

Low socioeconomic status is found to enhance the range of harmful risks from persistent contaminants. In this context and predominantly with respect to PCBs, the highest influence of socioeconomic and behavioral factors was observed in a coastal community of Chukotka in comparison to a more inland site. Individuals with low socioeconomic status had a 2–6 times higher risk of PCB exposure. The relatively high carcinogenic risks observed for residents of the two Chukotka communities in our study who consumed large quantities of marine mammal fat (i.e., those practicing traditional subsistence) seems unacceptable. Based on the current and previous studies [7,8], it is clear that action plans are needed to improve sanitation and to enhance the socioeconomic status of the indigenous population of Chukotka. Furthermore, it is recommended that individual risk assessment models include biomarkers of exposure to persistent contaminants.

Based on the current and previous studies [7,8], some action plans should be implemented to improve sanitation and enhance the socioeconomic status of the indigenous population of Chukotka. Furthermore, it is recommended that individual risk assessment models be based on biomarkers of exposure to persistent contaminants.

## Figures and Tables

**Table 1 ijerph-17-00128-t001:** Exposure pathway-specific health risks of polychlorinated biphenyls (PCBs) and DDTs (dichloro-diphenyl-trichloroethanes) calculated for a coastal indigenous population of Chukotka (the village of Uelen in 2001; *n* = 251 aged 18–71 year).

Exposure Pathways	Cancer Risk (CR)	CR Relative Contribution (%)	Noncarcinogenic Hazard (HQ)	HQ Relative Contribution (%)
**PCBs**				
Atmospheric air	4.8 × 10^−9^ [6.6 × 10^−10^−8.2 × 10^−9^]	<0.1	1.9 × 10^−5^ [1.4 × 10^−7^−3.2 × 10^−5^]	<0.1
Potable water (orally)	1.0 × 10^−7^ [4.5 × 10^−8^−2.2 × 10^−7^]	<0.1	0.0086 [0.0002–0.0133]	0.9
Household surfaces (scraping)	3.9 × 10^−5^ [1.8 × 10^−6^−7.7 × 10^−5^]	14. 5	0.07 [0.008–0.131]	7.2
Soil dust (orally and by inhalation)	3.1 × 10^−7^ [1.4 × 10^−8^−6.8 × 10^−7^]	0.1	0.055 [0.002–0.121]	6.0
Grocery store food	3.2 × 10^−6^ [2.4 × 10^−7^−6.0 × 10^−6^]	1.2	0.0025 [0.0002–0.0047]	0.3
Traditional food	2.3 × 10^−4^ [9.8 × 10^−6^−4.6 × 10^−4^]	84.2	0.78 [0.03–1.53]	85.6
Risk total	2.7 × 10^−4^ [1.2 × 10^−5^−5.4 × 10^−4^]	100	0.91 [0.04–1.78]	100
**DDTs**				
Atmospheric air	2.9 × 10^−9^ [4.5 × 10^−10^−5.2 × 10^−9^]	<0.1	6.8x10^−6^ [4.4 × 10^−7^–1.8 × 10^−5^]	0.3
Potable water (orally)	8.0 × 10^−7^ [7.8 × 10^−8^−1.4 × 10^−6^]	2.0	4.6 × 10^−5^ [6.8 × 10^−6^–8.4 × 10^−5^]	2.1
Household surfaces (scraping)	2.1 × 10^−5^ [4.0 × 10^−6^−3.9x10^−5^]	51.0	0.0012 [0.0002–0.0021]	50.5
oil (orally and by inhalation)	1.0 × 10^−10^ [9.2 × 10^−10^−1.8 × 10^−10^]	<0.1	4.0 × 10^−5^ [4.0 × 10^−6^–7.0 × 10^−5^]	1.7
Grocery store food	1.8 × 10^−7^ [2.4 × 10^−8^−2.9 × 10^−7^]	0.4	3.0 × 10^−5^ [2.0 × 10^−6^–8.0 × 10^−5^]	1.0
Traditional food	1.9 × 10^−5^ [2.0 × 10^−6^−3.5 × 10^−5^]	46.6	0.0010 [0.0001–0.0021]	44.4
Risk total	4.0 × 10^−5^ [6.0 × 10^−6^−7.5 × 10^−5^]	100	0.0024 [0.0003–0.0042]	100

**Table 2 ijerph-17-00128-t002:** Exposure pathway-specific health risks of PCBs estimated for a coastal indigenous population of Chukotka (the village of Uelen in 2010 *n* = 86 aged 21–67 year).

Health Risk	Cancer Risk (CR)	CR Relative Contribution %	Noncarcinogenic Hazard (HQ)	HQ Relative Contribution %
**PCBs**				
Atmospheric air	9,6 × 10^−10^ [1,2 × 10−11−1,8 × 10^−9^]	<0.1	8.0 × 10^−6^ [1.0 × 10^−7^–1.5 × 10^−5^]	<0.1
Potable water (orally)	2,0 × 10^−8^ [2,1 × 10^−9^−3,8 × 10^−8^]	<0.1	3.4 × 10^−5^ [4.0 × 10^−6^−6.4 × 10^−5^]	0.6
Household surfaces (scraping)	4,7 × 10^−5^ [7,2 × 10^−6^ −8,7 × 10^−5^]	21.16	0.078 [0.012–0.144]	5.4
Soil (orally and by inhalation)	7,1 × 10^−8^ [1,2 × 10^−8^−1,3 × 10^−7^]	0.1	5.3 × 10^−4^ [9.0 × 10^−5^−9.7 × 10^−4^]	3.8
Grocery store food	6,1 × 10^−6^ [3.8 × 10^−7^−7.4 × 10^−6^]	2.4	0.010 [0.0001–0.0027]	0.2
Traditional food	2.0 × 10^−4^ [1.5 × 10^−4^−3.3 × 10^−4^]	76.0	1.31 [0.97–2.21]	90.0
Risk total	2.4 × 10^−4^ [1.9 × 10^−4^−3.1 × 10^−4^]	100	1.44 [0.97–2.46]	100

**Table 3 ijerph-17-00128-t003:** Site and activity specific carcinogenic and noncarcinogenic health risks of PCBs, DDTs, mercury (Hg), and lead (Pb) for indigenous people of Chukotka in 2001.

Indicator	Value	Carcinogenic Risk (CR)	Noncarcinogenic Risk (HQ)
**PCBs (Aggregate of 15 Congeners)**			
Occupational activity, the village of Uelen (males only)	Traditional subsistence, *n* = 37	1.1 × 10^−3^ [9.3 × 10^−5^−2.0 × 10^−3^]	1.78 [0.16–3.40]
	Other occupations, *n* = 82	2.4 × 10^−4^ [2.1 × 10^−5^−4.6 × 10^−4^]	0.40 [0.04–0.77]
Remoteness from district centre	The village of Uelen (740 km), *n* = 251	2.3 × 10^−4^ [1.0 × 10^−5−^4.6 × 10^−4^]	0.39 [0.02–0.77]
	The village of Kanchalan (60 km), *n* = 360	3.8 × 10^−5^[0.0−1.0 × 10^−4^]	0.06 [0.0–0.17]
DDTs (Aggregate of DDT and its Four Metabolites)			
Occupational activity, the village of Uelen (males only)	Traditional subsistence, *n* = 37	2.4 × 10^−5^[7.3 × 10^−6^−4.0 × 10^−5^]	0.0014 [0.0004–0.0024]
	Other occupations, *n* = 82	1.9 × 10^−5^ [2.9 × 10^−6^−3.5 × 10^−5^]	0.0011 [0.0002–0.0020]
Remoteness from district centre	The village of Uelen (740 km), *n* = 251	1.9 × 10^−5^ [2.0 × 10^−6^−3.5 × 10^−5^]	0.0011 [0.0001–0.0021]
	The village of Kanchalan (60 km), *n* = 360	3.9 × 10^−6^[0.0−9.8 × 10^−6^]	0.0002 [0.0–0.0005]
Lead			
Occupational activity, the village of Uelen (males only)	Traditional subsistence, *n* = 37	2.4 × 10^−6^ [1.3 × 10^−6^ -3.6 × 10^−6^]	0.08 [0.03–0.14]
	Other occupations, *n* = 82	1.9 × 10^−6^ [6.5 × 10^−7^ -3.1 × 10^−6^]	0.06 [0.02–0.09]
Mercury			
Occupational activity, the village of Uelen (males only)	Traditional subsistence, *n* = 37	-	1.21 [0.66–1.76]
	Other occupations, *n* = 82	-	0.93 [0.35–1.51]
Remoteness from district center	The village of Uelen (740 km), *n* = 251	-	0.89 [0.31–1.47]
	The village of Kanchalan (60 km), *n* = 360	-	0.27 [0–0.68]

**Table 4 ijerph-17-00128-t004:** Calculated PCB-related cancer risks due to the consumption of traditional foods in relation to social variables (village of Uelen, 2010).

Indicator	Value	Carcinogenic Risk	Noncarcinogenic Risk
Monthly household income per capita	Below living wage (RUR 11,113.5), *n* = 33	3.4 × 10^−4^ [1.6 × 10^−4^−7.8 × 10^−4^]	0.57 [0.26–1.31]
	Above living wage, *n* = 9	1.0 × 10^−4^ [0.0−3.7 × 10^−4^]	0.17 [0.0–0.62]
Marine mammal fat consumption	Above 10 kg/year, *n* = 13	4.8 × 10^−4^ [2.5 × 10^−4^−1.0 × 10^−3^]	0.81 [0.42–1.72]
	Below 10 kg/year, *n* = 29	8.8 × 10^−5^ [0.0− 3.5 × 10^−4^]	0.15 [0.0–0.58]
Alcohol consumption	Low (up to 1 L of vodka monthly), *n* = 17	1.2 × 10^−4^ [8.0 × 10^−7^−4.0 × 10^−4^]	0.19 [0.0–0.66]
	High (2 L+ bottles of vodka monthly), *n* = 25	3.9 × 10^−4^[1.9 × 10^−4^−8.7 × 10^−-4^]	0.65 [0.32–1.45]
Occupational activity, the village of Uelen (males only)	Traditional subsistence, *n* = 6	5.8 × 10^−4^[3.2 × 10^−4^−1.2 × 10^−3^]	0.97 [0.53–2.00]
	Other occupations, *n* = 13	1.4 × 10^−4^ [1.8 × 10^−5^−4.4 × 10^−4^]	0.24 [0.03–0.74]
Awareness of persistent contaminants exposure prevention	Aware, *n* = 38	3.8 × 10^−4^[1.8 × 10^−4^−8.5 × 10^−4^]	0.63 [0.30–1.41]
	Unaware, *n* = 4	6.7 × 10^−5^ [0−3.1 × 10^−4^]	0.11 [0–0.52]

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
