# Peer review of "Health Risk Modifiers of Exposure to Persistent Pollutants among Indigenous Peoples of Chukotka"

_ijerph, 2019, doi:10.3390/ijerph17010128_

Round 1

Reviewer 1 Report

Pollution exposure is an important topic, but the field of pollution exposure  has become much more precise and technically advanced in recent years. The methods in this study are not well grounded in the scientific and statistical literature. The methods used a survey of food and other potential exposure routes to make a prediction about exposure risk of individuals. This work needs to be validated with some measurement of actual exposure measurement or a convincing link between this population and other research populations that did make measurements of exposure.

Also, the methods, results, and discussion are not clearly focused on this study. In fact, the conclusion moves to statements about sanitation and biomarkers when neither of these topics were included in any aspect of the study.

To make this research publishable, the quantification needs to be validated, and the connections from the methods to the results to the discussion need to be much more clearly developed.

Specific feedback on the text is below:

There are hundreds of grammatical errors that need to be rewritten. Numerous sentences are not written with straightforward construction. Please rewrite sentences to be more clear and remove the extra words in those sentences. Examples are below (but there are hundreds in the paper):

Abstract:

including PCBs, DDTs and lead, as well as mercury. Change to “including PCBs, DDTs, lead, and mercury” “Nearly 50% of past DDT-related health risks also appear to have been contribute by contaminated indoor surfaces involving commonly-used DDT-containing insecticides.” Change to: “Nearly 50% of past DDT-related health risks arise from contaminated indoor surfaces involving commonly-used DDT-containing insecticides.”

Objectives:

This is one of the most important sentences in the paper, but it does not make sense:

“The study aimed to assess the health risks among the indigenous people of Chukotka and focuses on pathway-specific exposures for the most common persistent contaminants in order to explore in some detail the impact of common socioeconomic and behavioral factors on cancer and non-cancer risks.”

Should be:

“The study aimed to assess the health risks among the indigenous people of Chukotka and to identify pathway-specific exposures for the most common persistent contaminants in order to better understand the impact of common socioeconomic and behavioral factors on cancer and non-cancer risks.”

Methods- again, the language is not clear:

“In order to quantify the non-carcinogenic risk of PCBs intake through atmospheric air, potable water (orally), household surface scrapings, soil dust (oral and inhaled), as well as traditional and grocery store foods, we used reference concentrations,…”

Is unclear. How were the reference concentrations used? How is the intake important for the quantification? How exactly was the quantification carried out?

“In addition to occupation, the influences of the following social status indicators were explored:”

This is a method discussion, so how were these indicators used in the study?

The methods section is lacking detail. The description should be enough to replicate the study.

How were the values in Table 1 and 2 determined? What is the method for calculating cancer risk? Where is the data that would lead to these calculations of cancer risk? Where is the measurement data? What assumptions were made in using the average PCB level in certain foods? Were the high and low levels also used in the calcuations or only the average values?

Discussion:

Some of the discussion points do not seem supported by any evidence. For example, “Lead exposure during the preparation of self-made lead pellets is another issue, as well as the inhalation of lead fumes released during gun use.” What is the amount of lead exposure when making pellets (this might be a high level) vs using a gun (which is likely a VERY LOW level). No quantitation of these values is discussed. They are just listed as maybe important.

Conclusions:

The conclusions are intended to arise from the study. However, comments in the conclusions such as:

 “Individuals with low socioeconomic status had a 2-6 times higher risk of PCB exposure.” Are not shown in the tables of data. How is this conclusion supported by results?

Rather suddenly, there is a switch from pollution to sanitation in the conclusion: “it is clear that action plans are needed to improve sanitation and to enhance the socioeconomic status of the indigenous population of Chukotka.” However, this was not a study of sanitation. How is this a relevant conclusion?

The last 2 sentences of the conclusion are repeated twice.

Reviewer 2 Report

The abstract of this work is very interesting but unfortunately the introduction, methods, and results section of the manuscript are lacking the specifics needed to support the discussion and conclusion of this work. There is limited information on the study population in the methods section or the overlap between the two study periods. Details of the questionnaire are also needed. Currently the data collected vs. the data from reference values is unclear and this needs to be corrected throughout the methods and results section. Overall the methods and data presentation do not provide enough information for peer review.

Introduction

The introduction could benefit from further context and citations. Currently it is a very brief overview that does not highlight the importance of studying the compounds of interest.

Citations are missing from the first few sentences

Methods

This section should be broken into subsections with headings for clarity for readers.

Results

It needs to be clearly stated where all values/equations are coming from and then the assumptions from those should be reviewed in the discussion. Is it not possible to use a high/medium/low estimate for intakes, etc. based on the questionnaire data?

Author Response

See attached file. Thank you for all valuable comments.

Round 2

Reviewer 1 Report

The paper is very much improved. The only remaining issue that I see is at the end of the conclusions:

"Based on the current and previous studies [7, 8], it is clear that action plans are needed to improve sanitation and enhance the socioeconomic status of the indigenous population of Chukotka."

This study was about Pb and Hg concentrations in blood that likely arose from dietary consumption and related activities. There is no specific investigation about sanitation or socioeconomic changes addressed in this research study. So, it seems to be an over-reach to state that "it is clear that action plans are needed....". The research does NOT show that action plans are needed, but just presents the problem quantitatively.

If this sentence is removed, then I don't have any further issues with the text.

Author Response

Response to review:

The mentioned sentence is changed. Thank you!

Reviewer 2 Report

Minimal changes from the rejected version have been made to the introduction and discussion. 

Author Response

I cannot find any specific advises, except a general evaluation. Thank you.